# First Record of Kemp’s Ridley Sea Turtle, *Lepidochelys kempii* (Garman, 1880), in the Waters of Martinique Island (Lesser Antilles)

**DOI:** 10.3390/ani14040596

**Published:** 2024-02-11

**Authors:** Damien Chevallier, Jacques Fretey, Jean Lescure, Marc Girondot

**Affiliations:** 1Unité de Recherche BOREA, MNHN, CNRS 8067, SU, IRD 207, UCN, UA, Station de Recherche Marine de Martinique, Quartier Degras, Petite Anse, 97217 Les Anses d’Arlet, France; 2Sea Turtle Research Center—Chélonée, 46260 Beauregard, France; jfretey@imatech.fr; 3National Museum of Natural History, Institute of Systematics, Evolution and Biodiversity–CP 30, 57 rue Cuvier, 75005 Paris, France; jean.lescure@mnhn.fr; 4Laboratoire Ecologie, Systématique et Évolution, Équipe de Processus Écologiques et Pressions Anthropiques, CNRS, AgroParisTech et Université Paris-Saclay, UMR 8079, 91190 Gif-sur-Yvette, France; marc.girondot@universite-paris-saclay.fr

**Keywords:** marine turtle, Kemp’s ridley, olive ridley, Lesser Antilles, distribution

## Abstract

**Simple Summary:**

The knowledge of the distribution of marine turtles is in perpetual change because of new data, but also due to the impact of climate change. We provide an update on the distribution of the two *Lepidochelys* species in the Caribbean Sea.

**Abstract:**

An adult female *Lepidochelys kempii* was found trapped in drifting sargassum south of Martinique; this is the southernmost report of this taxon in the Lesser Antilles arc. Determining the limits of distribution and the existence of possible sympatry between *L. kempii* and *L. olivacea* in certain subregions of the Caribbean has been hindered by numerous misidentifications. We review the available data and propose a new distribution map in the Caribbean, which can serve as a basis for future studies.

## 1. Introduction

Kemp’s ridley sea turtles, *Lepidochelys kempii* (Garman, 1880) [1], are the smallest sea turtles in the world. They are primarily found in the Gulf of Mexico [2], but juveniles are also found in the Atlantic Ocean as far north as Nova Scotia. The extension of the distribution of this species to the south of the Gulf of Mexico is not clearly known. For example, recent mapping of marine turtle regional management units limits this species to the Gulf of Mexico and east coast of the USA [3], ignoring its presence in the eastern North Atlantic [4,5].

When considering the unusual presence of the genus *Lepidochelys* in the Atlantic Ocean, we should first contextualize this observation in terms of its global distribution and verify the accuracy of records regarding the presence of each taxon in a particular region. Indeed, the recognition of two distinct taxa was rather late [6,7], and many environmental scientists still confuse the two species of the genus *Lepidochelys* and even sometimes *Lepidochelys* and *Caretta caretta*. For a long time, *Lepidochelys kempii* (Garman, 1880) was mistakenly thought to be a hybrid between the loggerhead sea turtle (*Caretta caretta*) and the hawksbill sea turtle (*Eretmochelys imbricata*) or green sea turtle (*Chelonia mydas*) [8]

According to Bowen et al. [9], mitochondrial DNA (mtDNA) analysis indicates that Kemp’s ridley and olive ridley sea turtles are distinct taxa in terms of matriarchal phylogeny (i.e., mtDNA inherited from the mother), although they form a monophyletic group distinct from other sea turtles. The separation of the two taxa may date back to the formation of the Isthmus of Panama and the closure of the gateway between the Pacific and Atlantic Oceans around 3–6 million years ago [10]. Molecular studies reveal two main genetic groups (J and K), each with a central haplotype (specific mtDNA sequences). The haplotypes from the K clade are found in *L. olivacea* in India and Sri Lanka, all of which present a characteristic sequence that is also found in *L. kempii* as well as other species of sea turtles. The K haplotype is therefore considered to be the ancestral haplotype. The J haplotype is found in *L. olivacea* across all basins of the Pacific and west Indian Oceans and is closely related to those of the east Pacific and Atlantic Oceans. Thus, *L. olivacea* with the J haplotype may be descended from the Indian and Sri Lankan populations, which served as sources for the Pacific and Atlantic populations [11].

The identification of *Lepidochelys* species based on scutes alone should be undertaken with caution. For example, the observer may rely on the presence of five pairs of symmetrical costals to identify *L. kempii*. Within a single nest of *L. olivacea*, however, newly hatched turtles can present an asymmetrical number of costals from five on one side to nine on the other [7,12,13,14], although some individuals can have five pairs of symmetrical costals [6,15] that resemble *L. kempii*.

Carr and Caldwell [16] examined the carapace patterns of 96 immature Kemp’s ridley sea turtles from the Florida coast. Only one did not have the usual five pairs of costals, instead showing five/six costals. Chávez et al. [17] counted the number of costal scutes on 154 adult females on Rancho Nuevo beach (Mexico). The following variations were observed: 5/5 (147 individuals); 5/6 (4); 6/5 (1); 6/6 (1); and 1 female with obscured sutures between her costals. In a histogram summarizing the number of costals observed in Kemp ridley sea turtles, Pritchard [7] noted the following patterns: 5/5 (296 individuals); 5/6 (5); 6/5 (2); and 6/6 (5). Counting the number of costals in three populations of *L. olivacea*, Pritchard reported five pairs of costals in 3 out of 743 adult turtles in Suriname, 3 out of 378 in Sri Lanka, and 14 out of 102 in Honduras. The same author also mentioned that 26 out of 116 olive ridley sea turtles (22.4%) had five pairs of costals on Isla de Ratones (Puerto Rico). These different examples highlight that the identification of *L. kempii* based on the number of costals is too restrictive.

The decomposed carcass of an individual without a head which could belong to the *L. kempii* species was found in 2013 in Martinique (Marcel Bourgade, unpublished report), but the quality of photography led to some doubts about its identification. In this study, we provide new evidence for the presence of Kemp’s ridley turtle in the Lesser Antilles in the Caribbean Sea, and we provide an update on the distribution of the two *Lepidochelys* species in this region. The new observation reported here is the first certified observation of an alive specimen of *Lepidochelys kempii* in Martinique.

## 2. Materials and Methods

On 8 April 2021, three yachtsmen observed a motionless sea turtle trapped in rafts of sargassum floating southeast of Martinique between Ilet Chevalier and Cap Chevalier (14°25′56.2″ N/60°49′36.8″ W). Since 2011, the pelagic algae *Sargassum fluitanse* and *Sargassum natans* have periodically invaded the Caribbean waters and coasts on a massive scale, with intensification since 2017. The algae may not increase the energetic costs of nesting after a turtle has emerged onto the beach, but costs may be imposed in algae-filled waters as turtles initially seek to emerge [18].

The turtle was hauled into the boat and brought back to land. One of us (Damien Chevallier) immediately went to examine it and observed its weak state.

The identification of the turtle by observing its morphological features left no doubt: it was a Kemp’s ridley sea turtle, *Lepidochelys kempii* (Garman, 1880) [1] (Figure 1, Figure 2 and Figure 3). The curved carapace length (CCL) of this adult female was 62 cm. The amputated left foreleg was reduced to a scarred stump. No other visible wounds were observed.

The Kemp’s ridley sea turtle was released to Grande Anse des Salines (Saint-Anne) in an area free of sargassum on the same day. The turtle initially seemed to have trouble swimming, but, despite its stump, its swimming became easier as it swam deeper.

Regarding the pattern of the head scutes, two frontonasals were separated from the frontal by what may be called two prefrontal supraoculars connected by a short, curved, oblique suture. Three postoculars were apparent on each side. On the right profile, the first superior postocular was separated from a large supraocular incompletely divided into an S shape by a small oval subocular. A transverse loreal on each side connected the supraciliaries to the prefrontals.

On the plastron, there were two tiny asymmetrical triangular gulars and four inframarginals. There was also a short, almost non-existent suture at the junction between the abdominals, femorals, and inframarginals 3 and 4 (Figure 3), as well as three small anal margins.

## 3. Discussion

### 3.1. Caribbean Distribution of the Two Taxa

*L. kempii* is only found in the Atlantic Ocean, mostly in the temperate waters of the north Atlantic up to Canadian and European waters (including the Mediterranean Sea) as well as in the Gulf of Mexico. Until now, its known southern limit did not go beyond the line joining the north of Venezuela and Cuba in the west Atlantic Ocean to Madeira in the east [19,20]. Nesting was reported in Colombia by Chávez and Kaufmann [21].

The southern localization of *L. kempii* in Venezuela was erroneously reported by Donoso-Barros [22] and Flores [23] around Cumaná (Edo. Sucre), La Tortuga Island, and Piritú Island (Edo. Anzoátegui). Pritchard and Trebbau [24] later confirmed that this observation was incorrect and that these individuals were instead *L. olivacea*. They mentioned 29 adult female olive ridley sea turtles ringed in Eilanti (Suriname) and recaptured in Venezuela and Trinidad. Bacon [25] described the oviposition of the species on Matura and Manzanilla beaches in Trinidad, but *L. olivacea* is not known to nest in Venezuela [26].

In the Lesser Antilles arc, Aguayo [27] mentioned *L. kempii* in Cuba, but after examining the preserved specimen, Carr [6] identified it as *L. olivacea*. Bacon [25] reaffirmed the presence of *L. kempii* in Cuba. In their Table 3.2, Azanza Ricardo et al. [28] listed *L. kempii*, probably by error, as nesting in Cuba, but without further details, this information cannot be confirmed. Dunn [29] mentioned *L. kempii* in Jamaica, but this was later rejected by Lewis [30].

*L. olivacea* has a circumglobal distribution and occupies a more southern range than *L. kempii*. Our knowledge about the northern limit of its distribution range in the Atlantic Ocean is rather imprecise. To the east in northern West Africa, *L. olivacea* has been observed in the Cape Verde archipelago [31], Senegal [32], and Mauritania [6,33].

Bacon [25] mentioned possible the foraging habitats of *L. olivacea* along the Panama coast. Its presence in Venezuela is occasionally highlighted. An apparently reliable observation of the species was made off Vieques (Puerto Rico) by Caldwell and Erdman [34]. Bacon [25] referred to its foraging habitats around Vieques and the nearby US Virgin Islands, which should be confirmed by new observations. The author mentioned the occasional foraging of *L. olivacea* in Bermuda, which seems more doubtful, and noted the unconfirmed presence of the species in the Dominican Republic. Tufts [35] reported a *Lepidochelys* captured near Cartagena de Indias in the collections of the Instituto La Salle de Bogotá (Colombia). This specimen, described and illustrated by Nicéforo María [36], was identified as *L. kempii*. Pritchard and Trebbau [24] later referred to this individual; in their view, its six pairs of costal scutes and its carapace width measuring 88.49% of its length clearly point to *L. olivacea*. In 2002, the Colombian Ministry of the Environment categorized six species of sea turtles present in the waters and coasts of Caribbean Colombia for nesting or foraging. Among them, the Ministry cites the “accidental” presence of the Golfina turtle (*L. olivacea*) and the Lora turtle (*L. kempii*). However, in his inventory of nesting and foraging areas in Caribbean Colombia, Ceballos-Fonseca [37] mentioned neither taxon.

### 3.2. Lepidochelys in the French West Indies

Father Pinchon [38] distinguished two types of loggerhead sea turtles in the West Indies, including *Caretta kempii* (Germare), which, in his view, came to the Caribbean Sea from the Gulf of Mexico. Kermarrec [39] noted that the species *L. olivacea* is uncommon in the Caribbean islands and that Venezuela and Trinidad are marginal areas.

For Dropsy [40], the “Zekal ronde” mentioned by fishermen could be *L. kempii*, although Lescure [41] later considered it more likely to be *L. olivacea*. This identification was confirmed after examining the carapace of an immature *L. olivacea* from the collection of the Father Pinchon Museum in Fort-de-France [41].

Bacon [25] refers to a foraging site for *L. olivacea* in Martinique. Anne and Peter Meylan were informed about a young olive ridley sea turtle held in captivity in Case Pilote that had been captured in the nearby area [42,43].

Carr, Meylan, Mortimer, Bjorndal, and Carr [42] as well as Fretey and Lescure [44], mention a species of the genus *Lepidochelys* in the Guadeloupe archipelago. Meylan [43] discovered three carapaces and a skull of *L. olivacea* of unknown origin at a shop in Basse-Terre, while Fretey viewed a painted carapace of *L. olivacea* in a restaurant in Saint-François in 1983 (not published). Four adults of *L. olivacea* were reported in Guadeloupe by Fretey and Lescure [45] and Cuvillier in [46]: off the coast of Ilets Pigeon (20 January 1998; CCL: 68 cm); in Ilet Boissard (female; 20 October 1998; CCL: 65.5 cm); and off the coast of Banc des Flandres (female; 2 January 1999; CCL: 57 cm). A fourth individual was treated at the Aquarium of Guadeloupe in Le Gosier after ingesting oil (Figure 4). No individual of the genus *Lepidochelys* was reported in the waters of Saint-Barthelemy and Saint-Martin [45,47] until the accidental capture of a *Lepidochelys* in a net at the site known as the Washing Machine at the peak of the Morne de Grand Fond.

Dow et al. [48] as well as Eckert and Eckert [49] list 41 nesting sites for Kemp’s ridley sea turtles and 62 for olive ridley sea turtles in the Caribbean region (Figure 5).

## 4. Conclusions

The precise limits of the distribution range of the two taxa are still very uncertain. Accurate data are rare due to the numerous identification problems and the labelling errors between *L. kempii*, *L. olivacea*, and *C. caretta* in museums, not to mention the lack of interest among environmental scientists to observe turtles at sea compared to nesting sites. However, the reliable data point to possible sympatry of the two taxa—and their possible hybridization—in the north of the Lesser Antilles arc. Indeed, the molecular analyses of Barber et al. [50] and Pfaller et al. [51] confirmed cases of natural hybridization between *L. kempii* and *C. caretta*. Hybridization work has already entered the genomic era, and the new data will certainly revolutionize this field with more evidence of old and frequent hybridizations [52].

Despite being recently described in Indonesian waters [53], Kemp’s ridley’s presence outside of the Atlantic basin is doubtful: Donna Shaver (Division of Sea Turtle Science and Recovery, Padre Island National Seashore, Corpus Christi, Texas) wrote the following on 12 December 2023 about the presence of this species in the Indian Ocean in the Cturtles mailing list: “I agree with Nathan [J. Robinson] and Matt’s [Warre] skepticism that this turtle was a Kemp’s. Based on geographic distribution and the highly variable costal scute counts of olive ridleys, this was likely an olive ridley and not a Kemp’s ridley.” Any description of a species outside its known range should be accompanied, if possible, by a genetic identification. This was not possible for the *L. kempii* found in Martinique, as we preferred to release the animal as quickly as possible.

In the current context of climate and oceanographic change, an increase in the distribution range of a sea turtle species should be carefully monitored, as it may lead to the colonization of new geographic areas. There may be a tendency to underestimate the importance of discovering an individual outside of its usual distribution range, and thus, view it as an anomaly. However, this individual, especially if it is a fertilized female, may explain the observation of a new nesting colony.

## Figures and Tables

**Figure 1 animals-14-00596-f001:**
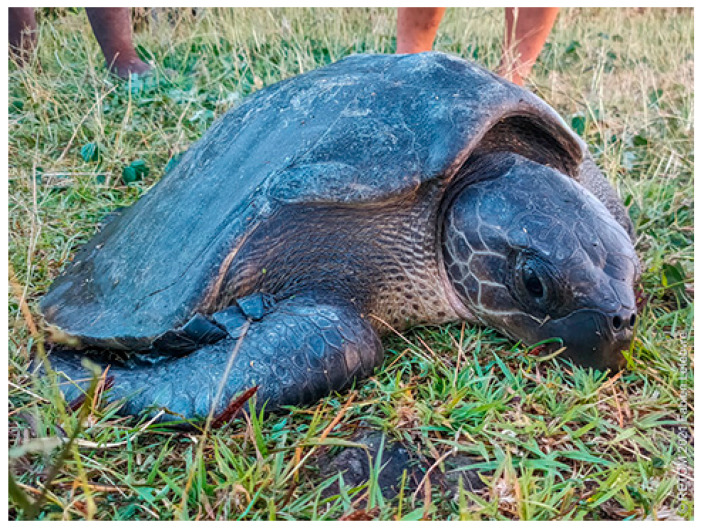
Anterior view of Kemp’s ridley sea turtle discovered near Ilet Chevalier in Martinique on 8 April 2021 (photography by Fabien Lefebvre/CNRS).

**Figure 2 animals-14-00596-f002:**
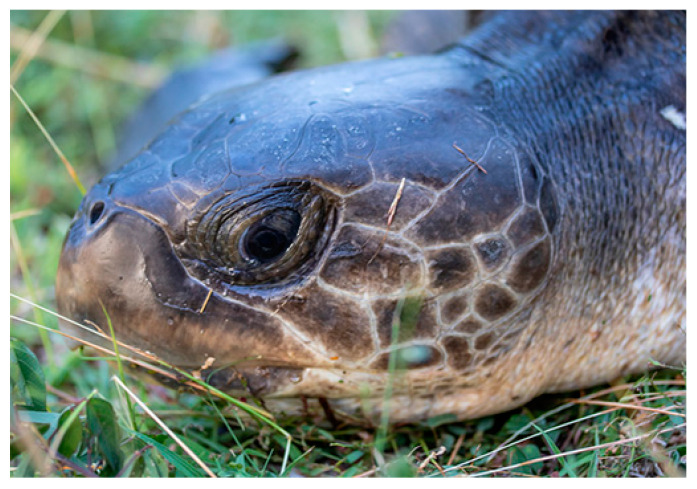
Left profile of the head scutes of Kemp’s ridley sea turtle discovered near Ilet Chevalier in Martinique on 8 April 2021 (photography by Fabien Lefebvre/CNRS).

**Figure 3 animals-14-00596-f003:**
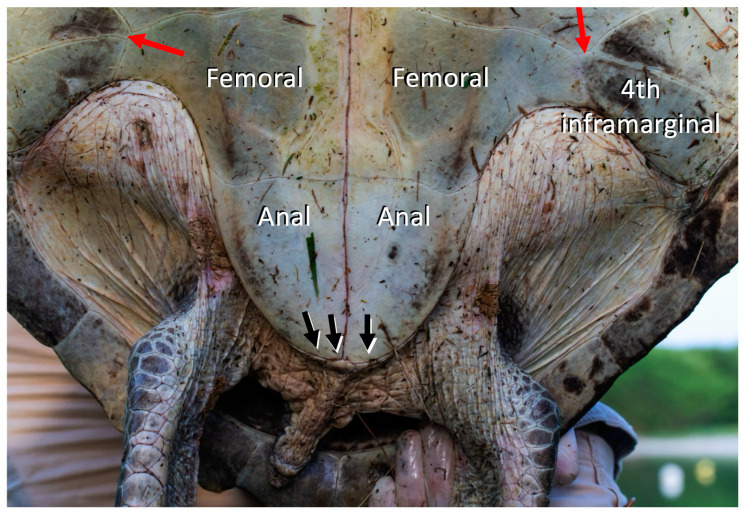
Detail of the posterior part of the plastron of Kemp’s ridley sea turtle discovered near Ilet Chevalier in Martinique on 8 April 2021 (from the photograph of Fabien Lefebvre/CNRS). Red arrows: sutures used for identification purposes. Black arrows: anal margins.

**Figure 4 animals-14-00596-f004:**
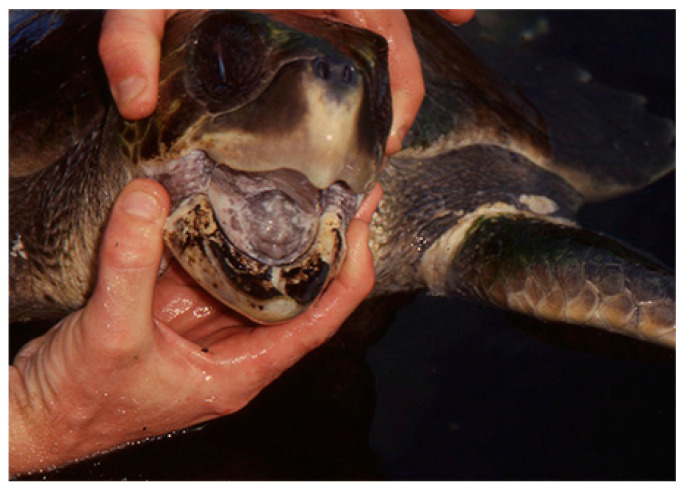
Individual of the species *L. olivacea* rescued off the Guadeloupe coast after being caught in an oil spill. Traces of oil are still visible on the lower beak (photography by J. Fretey).

**Figure 5 animals-14-00596-f005:**
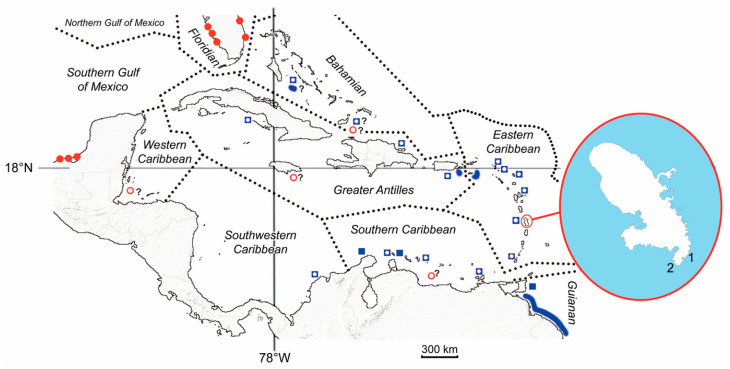
Cartographic representation of our knowledge on the presence of the genus *Lepidochelys* in the Caribbean region. *L. olivacea*—open blue square: reported presence; solid blue square: known nesting site; ovoid shape: foraging site. *L. kempii*—open red circle: reported presence; solid red circle: known nesting site. Magnifying glass: localization of beaching (1) and release (2) of *L. kempii* in Martinique.

## Data Availability

Other photographs are available under request to the correspondent author.

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
