# Peer review of "First Record of Kemp’s Ridley Sea Turtle, Lepidochelys kempii (Garman, 1880), in the Waters of Martinique Island (Lesser Antilles)"

_animals, 2024, doi:10.3390/ani14040596_

Round 1

Reviewer 1 Report

Comments and Suggestions for Authors

I really enjoy the note. I believe this is a good record and opens several questions about the distribution and behavior of this species.

Author Response

Reviewer 1

Line 39-40: Is this representing a problem for sea turtles and other marine species? you can mention that.
We have added this precision:

The algae may not increase the energetic costs of nesting after a turtle has emerged onto the beach, but costs may be imposed in algae-filled waters as turtles initially seek to emerge [6].

  1. Maurer, A.S.; Stapleton, S.P.; Layman, C.A.; Burford Reiskind, M.O. The Atlantic Sargassum invasion impedes beach access for nesting sea turtles. Climate Change Ecology 2021, 2, doi:10.1016/j.ecochg.2021.100034.

Line 42: Damien Chevallier?

DC has been changed to Damien Chevallier.

Line 48 : When was released?

This precision is added:

The Kemp’s ridley sea turtle was released to Grande Anse des Salines (Saint-Anne) in an area free of sargassum on the same day.

Line 64 : The first Kemp's ridley documented? is not clear. It seems like you are talking about the same turtle mentioned previously. Also, I think it will sound better if you talk about this register before.

We have reorganized the paper based on recommendation of the third referee and this precision is added before the new description as requested.

Line 128: nest? I know the turtle is present in the feeding ground areas but I am not sure about nesting in the country.
The referee is perfectly right. We have corrected:

Bacon [25] described the oviposition of the species on Matura and Manzanilla beaches in Trinidad but L. olivacea is not known to nest in Venezuela [26].

  1. Barrios-Garrido, H.; Arias-Ortiz, A.; Balladares, C.; Espinoza-Rodríguez, N.; García-Cruz, M.; González, M.F.; Pulido-Petit, G.; Rojas-Cañizales, D.; Rondon-Medicci, M.; Vernet, P.; et al. Records of olive ridley marine turtles (Lepidochelys olivacea Eschscholtz 1829) in Venezuelan waters: A review of historical data sets and threats. Chelonian Conservation and Biology 2022, 22, doi:10.2744/CCB-1514.1.

Line 211: This is great!!!

Thanks a lot!

Reviewer 2 Report

Comments and Suggestions for Authors

Although this work shows the possibility of having found specimens of L. kempii within the limits described for these species, it is always important to describe new findings, which would mean that the species is rapidly increasing its population and that its distribution continues to increase. They would be found in different areas previously unthinkable. However, care must be taken and confirmed by molecular biology to establish identity correctly. In the discussion, it is left to be seen that there may be the possibility that it could be misdescribed due to its morphology.

Line 45. Although figures of L. kempii are shown, it is suggested to add current figures of the individual found.

Line 66. Again, although the description is based on Figures 1 to 3, no figures are shown for the individual currently described. It is suggested not only to describe the well-described morphology but also to observe that comparison with the individual found. Describe. 

Line 117. Although all the background information is described and the quotes refer to the comparisons already made on L. kempii vs. L. olivacea, nothing new is added to what was previously described. Therefore, it is suggested to rewrite this section, especially in the comparison of the traits that the authors want to establish as the differences between these two species in particular. Because they later describe facts that would suggest that hybrids of L. kempii and C. caretta exist.

Line 192. In the area of molecular biology, this reference is no longer recent. I suggest you also look at something more recent DOI: 10.3390/d14050390. What do you suggest for future identification and confirmation of this species or hybrids? Explain

Author Response

Reviewer 2

Although this work shows the possibility of having found specimens of L. kempii within the limits described for these species, it is always important to describe new findings, which would mean that the species is rapidly increasing its population and that its distribution continues to increase. They would be found in different areas previously unthinkable. However, care must be taken and confirmed by molecular biology to establish identity correctly. In the discussion, it is left to be seen that there may be the possibility that it could be misdescribed due to its morphology.

We add this precision in discussion:
Any description of a species outside its known range should be accompanied, if possible, by a genetic identification. This was not possible for L. kempii found in Martinique, as we preferred to release the animal as quickly as possible.

Line 45. Although figures of L. kempii are shown, it is suggested to add current figures of the individual found.

The figures 1 to 3 are the original photography of the L. kempii found in Martinique. It is described in the legend of the figures:

Figure 1. Anterior view of Kemp’s ridley sea turtle discovered near Ilet Chevalier in Martinique the April 8th, 2021 (Photography by Fabien Lefebvre/CNRS).

Figure 2. Left profile of the head scutes of Kemp’s ridley sea turtle discovered near Ilet Chevalier in Martinique, the April 8th, 2021 (Photography by Fabien Lefebvre/CNRS).

Figure 3. Detail of the posterior part of the plastron of Kemp’s ridley sea turtle discovered near Ilet Chevalier in Martinique, the April 8th, 2021 (from the photograph of Fabien Lefebvre/CNRS). Red arrows: sutures used for identification purposes. Black arrows: anal margins.

Line 66. Again, although the description is based on Figures 1 to 3, no figures are shown for the individual currently described. It is suggested not only to describe the well-described morphology but also to observe that comparison with the individual found. Describe.
See previous answer.

Line 117. Although all the background information is described and the quotes refer to the comparisons already made on L. kempii vs. L. olivacea, nothing new is added to what was previously described. Therefore, it is suggested to rewrite this section, especially in the comparison of the traits that the authors want to establish as the differences between these two species in particular. Because they later describe facts that would suggest that hybrids of L. kempii and C. caretta exist.
As this section has been moved, we think that the request of the referee is no longer required.

Line 192. In the area of molecular biology, this reference is no longer recent. I suggest you also look at something more recent DOI: 10.3390/d14050390. What do you suggest for future identification and confirmation of this species or hybrids? Explain
We agree that 2003 is not recent in the context of molecular biology, but the result obtained by Barber et al (2003) is still valid. We remove the “recent” term.

The work about hybridization will benefit from the genomic era. We add this precision and this reference:

Hybridization work has already entered the genomic era, and the new data will certainly revolutionize this field with more evidence of old and frequent hybridizations [55].

  1. Vilaca, S.T.; Piccinno, R.; Rota-Stabelli, O.; Gabrielli, M.; Benazzo, A.; Matschiner, M.; Soares, L.S.; Bolten, A.B.; Bjorndal, K.A.; Bertorelle, G. Divergence and hybridization in sea turtles: Inferences from genome data show evidence of ancient gene flow between species. Molecular Ecology 2021, 30, 6178-6192, doi:10.1111/mec.16113.

Camacho-Sánchez et al. (2022) describe new haplotype for L. kempii. It will not help to describe hybrids. We do not add this reference.

Camacho-Sánchez FY, Narváez-Zapata JA, Acosta-Sánchez HH, López-Hernández M, Luzariaga-Neira A, Rodríguez-González H, Aguirre AA, Reyes-López MA (2022) Molecular identification and novel mitochondrial COI gene haplotypes of nesting Kemp’s ridley turtles (Lepidochelys kempii) in Rancho Nuevo Sanctuary, Mexico. Diversity 14

Reviewer 3 Report

Comments and Suggestions for Authors

The paper: "First record of Kemp’s ridley sea turtle, Lepidochelys kempii  (Garman, 1880) in the waters of Martinique Island (Lesser Antilles)" is an interesting study which presents the first finding of Kemp's ridley sea turtle in the area and maps distribution of two Lepidochelys species in the Caribbean. 

In great detail, the authors elaborate on their identification of the species (and pitfalls of possible confusion with Lepidochelys olivacea). However, I feel that the paper would benefit more from being reorganized in a typical scientific paper structure, with an Introduction where both species and their known ranges as well as the history of the research in the region would be elaborated, a Materials and Methods section, where the authors would describe sources of their data (from literature, unpublished field records, etc.) and criteria for species identification, and Results where the findings would be presented. Discussion and Conclusions are already present and I see no issues with those sections. Sea turtles are among the most vulnerable animal taxa today and any published bit of knowledge on their distribution might have important implications for future conservation efforts. Therefore, I would recommend this paper for publishing in Animals after a minor revision, related mostly to the reorganization of the paper. I have some additional comments in the annotated .pdf file. 

Author Response

Reviewer 3

The paper: "First record of Kemp’s ridley sea turtle, Lepidochelys kempii  (Garman, 1880) in the waters of Martinique Island (Lesser Antilles)" is an interesting study which presents the first finding of Kemp's ridley sea turtle in the area and maps distribution of two Lepidochelys species in the Caribbean. 

In great detail, the authors elaborate on their identification of the species (and pitfalls of possible confusion with Lepidochelys olivacea). However, I feel that the paper would benefit more from being reorganized in a typical scientific paper structure, with an Introduction where both species and their known ranges as well as the history of the research in the region would be elaborated, a Materials and Methods section, where the authors would describe sources of their data (from literature, unpublished field records, etc.) and criteria for species identification, and Results where the findings would be presented. Discussion and Conclusions are already present and I see no issues with those sections. Sea turtles are among the most vulnerable animal taxa today and any published bit of knowledge on their distribution might have important implications for future conservation efforts. Therefore, I would recommend this paper for publishing in Animals after a minor revision, related mostly to the reorganization of the paper. I have some additional comments in the annotated .pdf file. 

We agree we the recommendation of the referee and therefore the category of the paper has been changed to “Article”.

Line 20: Italicize species names.

Done

Line 36: I understand that this is a specific paper dealing with a specific finding and mapping of sea turtles, but I thing it would benefit from being organized in a more conservative manner, i.e. consisting of Introduction, Materials and Methods (where the authors would discuss the origins of the data - literature and unpublished field reports, etc.), Results (where the findings would be given out in detail),  Discussion and Conclusions (which are already present).

Done.

Line 44: It would be fine if the diagnostic characters would be pointed on figures by arrows.

Done:

Line 73: Is this the photograph of the same specimen or an example photo from other source? Be specific.

Also, I would prefer the text on the figure to be in English, because the paper is written in that language.

The precision is added, and language change has been done.

Line 73: This should go partly to Introduction and partly to M&M.

Done.

Line 92-94: Italicize

Done

Line 117 : This passage is also for the Introduction section, or to be included in Discussion.

Done.

Line 137: circumglobal

Done

Line 157 : Also, more fit for introduction or Discussion.

Done.

Line 191: I wouldn't use inevitable, although the hybridization is proven it is still a strong statement.

We agree; we change “inevitable” to “possible”.

However, the reliable data point to a possible sympatry of the two taxa – and their possible hybridization – in the north of the Lesser Antilles arc.